# Progress on In Situ and Operando X-ray Imaging of Solidification Processes

**DOI:** 10.3390/ma14092374

**Published:** 2021-05-02

**Authors:** Shyamprasad Karagadde, Chu Lun Alex Leung, Peter D. Lee

**Affiliations:** 1Department of Mechanical Engineering, Indian Institute of Technology Bombay, Mumbai 400076, India; 2Department of Mechanical Engineering, University College London, London WC1E 7JE, UK; alex.leung@ucl.ac.uk (C.L.A.L.); peter.lee@ucl.ac.uk (P.D.L.); 3Research Complex at Harwell, Harwell Campus, Oxfordshire OX11 0FA, UK

**Keywords:** solidification, microstructure evolution, X-ray imaging, semi-solid, additive manufacturing

## Abstract

In this review, we present an overview of significant developments in the field of in situ and operando (ISO) X-ray imaging of solidification processes. The objective of this review is to emphasize the key challenges in developing and performing in situ X-ray imaging of solidification processes, as well as to highlight important contributions that have significantly advanced the understanding of various mechanisms pertaining to microstructural evolution, defects, and semi-solid deformation of metallic alloy systems. Likewise, some of the process modifications such as electromagnetic and ultra-sound melt treatments have also been described. Finally, a discussion on the recent breakthroughs in the emerging technology of additive manufacturing, and the challenges thereof, are presented.

## 1. Introduction

A vast majority of engineered metallic components share two common initial processing steps: the melting and solidification of an alloy into granules, ingot, or near-net shape components. Understand the process-structure-property relationship is key to producing end-products with a desirable performance. The solidification of alloys is governed by the following sub-processes: nucleation and growth, precipitation, phase transformations, and mechanics at the macroscale. During solidification, metallic systems would undergo changes across time (nanoseconds to hours) and length scales (nanometres to metres), and every change can impact the mechanical, electrical and thermophysical properties [1]. For example, a typical automotive connecting rod can be produced using one of many processing routes, such as casting, forging, semi-solid forming, powder metallurgy (PM), and additive manufacturing (AM) methods [2,3,4,5]. Each processing route involves, at some stage, heating of the alloy to high temperatures (200–1600 °C), cooling with rates over a range of 1 to 10^6^ K/s, pressurization and mechanical deformation under strain rates of 1 to 10^4^ s^−1^ and loads of 1 to 10^5^ N [1,6]. The process window defined by these parameters plays a critical role in controlling the final microstructure, chemical homogeneity, grain size and orientation (texture), precipitates, and defects that influence the properties of the component. 

Solidification is an integral part of many processing routes, producing the initial chemistry and microstructures that set the initial phases and their nano-, micro- and macroscopic distributions. Therefore, understanding the approaches to design and control solidifying microstructures, including defects, is crucial in developing new alloys and processes, as well as further optimisation of the established routes. A primary disadvantage of the conventional post-mortem microstructural analysis is that the dynamics of microstructural evolution and the role of the surrounding regions are seldom captured. Therefore, in situ visualizations of solidification processes, ideally in three-dimensions, is one of the most desired approaches as it allows for real-time visualisation of the processes at multiple length and time scales.

However, the challenges associated with the real-time imaging of metallic systems often involve heating to extremely high temperatures, sensitivity to the surrounding atmosphere (pressure and reactive conditions), and more importantly, devising ways to look through a 3D specimen. To date, the bulk of the studies that reported material microstructures are based on post-mortem analyses [7,8]. However, with the establishment of synchrotron radiation facilities worldwide having high-flux and stable X-rays (1–10^5^ keV), many unprecedented scientific findings have been made possible over the last two decades [7,9,10,11,12,13]. The static 3D internal information of a specimen is now routinely obtained using synchrotron or laboratory tomography. However, capturing dynamically evolving changes (in seconds to milliseconds) with millimeter to sub-micrometer resolution requires specialised experimental equipment, often only available at a handful of synchrotrons in the world [14]. Furthermore, it is now also possible to obtain crystallographic and chemical information of metallic specimens at the nanometre resolution using a diverse set of X-ray based synchrotron characterisation methods including: diffraction to obtain bulk atomic measurements, phases, crystal defects and strains [13], and spectroscopy to probe chemical constituents and interfacial structures [15]. Figure 1 shows an overview of different synchrotron techniques used for probing information on metallic systems. 

During the late 1960s and early 70s, the first X-ray radiography experiments were performed to quantify how solidification interfaces evolved in metallic systems (e.g., [16], see Section 3.1 below for a detailed description). Several measurements established a theoretical understanding of the solute distribution and time-scales of dendritic growth in metals and demonstrated the potential of real-time observations that provided data for validating theories, numerical models [17,18,19,20], and shed new insights into a wide range of solidification mechanisms. These insights were mainly in 2D due to the limitation of detector technology to convert X-ray into visible light for fast-speed imaging, therefore most studies employed thin specimens that restricted the information in the 3rd dimension. The early years of the 21st century witnessed a paradigm shift in in situ experimentation, first in terms of radiographic capture in tens of milliseconds, and resolutions of few micrometres; and then via 3D tomographic scans of solidifying and deforming specimens acquired in tens of seconds, whose sizes ranged over a few millimetres in diameters (e.g., [21,22,23,24], see below). This has been made possible by the establishment of about a dozen 3rd generation synchrotrons, amongst a total of about 50 worldwide, such as the Advanced Photon Source (APS, Lemont, IL, USA), European Synchrotron Radiation Facility (ESRF, Grenoble, France), Swiss Light Source (SLS, PSI, Villigen, Switzerland), Diamond Light Source (DLS, Harwell, UK), and Spring-8 (Hyogo, Japan). The in situ cells for such experiments have several complexities and challenges, particularly in developing precision heating, motion and rotation mechanisms, and integration with beamline control for specific needs. Over the last few years, the focus has been on exploring even more challenging processes, such as AM [25,26,27], high-pressure die casting (HPDC) [28,29], and inertial welding [30] where the cooling rates are over 10^3^ K/s, and requires nearly sub-micron spatial resolutions, thus requiring frame rates of about 10^6^ Hz. 

The primary objective of this article is to review a wide range of feasibilities of performing in situ X-ray imaging experiments for metals processing involving solidification, and semi-solid deformation, with an emphasis on capturing the microstructural evolution and its role on the formation of defects. Several key observations have helped develop analytical and empirical models of nucleation and growth of microstructures, with an accurate description of the morphological parameters and three-dimensional features (e.g., [23,31], see below). Likewise, the in situ and operando (ISO) studies showed that, while defects such as intermetallics (such as Fe-intermetallics in secondary aluminium) and gas porosity evolve along with the microstructures ([32,33]), the hot-tears/hot cracks, and shear bands are strongly governed by the microstructural response to the hydrodynamical and mechanical deformation, in addition to those affected by the transport phenomena during solidification (e.g., [34,35], see below). During solidification, mechanical deformation is known to originate from imposed loading as well as weak thermal contractions, that are sufficient to cause hot-tearing (e.g., [36]). The dilation of the intergranular regions during deformation is known as Reynold’s dilatancy, and such dilatant shear bands accommodate hot-tears and shrinkage porosity (e.g., [29]). Various microstructures that form during solidification define the characteristics of the semi-solid region, where freely growing grains lead to granular behaviour and provide unintuitive responses to weak loads. Such behaviours, in metals, can only be captured with the help of in situ 3D imaging, which requires integration of a thermo-mechanical loading environment around the sample while capturing tomographic images (e.g., [37]). The mechanisms of defect formation can only be fully understood by capturing the complete phenomena and dynamics at the appropriate spatial-temporal resolution. Another specific need for real-time observations is to validate theories and classical hypotheses of solidification science. As a result, several studies have reported the quantification of microstructures that evolve during solidification, defects such as porosity and hot-tears, with the latter requiring deformation of semi-solid specimens. Numerous examples of how such observation in 3D plus time have led to the design of new alloys [38], new processes [39] as well as improvisation of existing technologies [13]. 

Prior review articles provide an overview of synchrotron X-ray imaging of metals [10,38,40,41]. This article aims to provide a comprehensive review of in situ and operando environment cell techniques, as well as describe new applications in emerging areas of flexible materials processing technologies such as AM [9,12,42]. A primary motivation for including ISO techniques is to present key advances in ISO cells, and to make researchers aware of experimental challenges when using ISO cells, so that experimental tasks are clearly planned and executed, ensuring effective utilisation of synchrotron beamtimes. The outline of the review article is as follows. We first present a review of in situ X-ray imaging techniques for metals, and then for ISO cells. Subsequently, an overview of key advances in observations of solidification microstructures, formation of defects, and the response to deformation is discussed. Finally, a brief overview of the current and future trends in emerging areas such as additive manufacturing is presented, where X-ray imaging studies have made remarkable contributions. The concluding sections also highlight the upcoming capabilities in terms of imaging as well as ISO techniques.

## 2. A Brief Review of Techniques

### 2.1. An Overview of X-ray Imaging Techniques for Metals Processing

The most popular method of obtaining 2D and 3D in situ microstructural information in metals is by absorption contrast X-ray imaging. The darker regions of the images (fewer X-rays reaching the detector) represent a more attenuating phase, i.e., a region of higher electron density, which increases with increasing atomic number (Z) and physical density; while the lighter regions represent less dense, lower Z regions. In order to obtain three-dimensional tomographic reconstructions, a series of radiographs are collected and then reconstructed in 3D using filtered back projection or iterative reconstruction techniques [43,44,45,46], which normally assumes the features to be static during acquisition. In cases where distinct phases in the materials have similar attenuation characteristics, phase contrast imaging techniques can be used which are up to 1000× more sensitive, and are based on the shift in phase that occurs as the X-rays travel at slightly different speeds, and can be captured as constructive or destructive interference on a detector placed at different distances from the specimen [11,43,47]. However, this technique is less suited for fast real-time capture of evolving features. On most synchrotron imaging beamlines, 3D tomographic reconstruction algorithms are built-in to the acquisition software, e.g., DAWN [48], SAVU [49], PyHST2 [50], TomoPy [51], etc.

### 2.2. Overview of In Situ and Operando (ISO) Environmental Cell Techniques

A considerable number of studies have effectively integrated ISO cells in custom-built X-ray imaging units and commercial X-ray computed tomography machines, that are now capable of resolving sub-micron features for tomographic capture. However, the following key limitations continue to exist when compared with synchrotron based units, i.e., (i) slower acquisition times (by about 100 times) for the same spatial resolution, (ii) loss of spatial resolution due to positioning of the ISO cell is further away from the X-ray source, and (iii) significantly less working space inside the X-ray imaging unit for mounting the ISO cells, restricting certain capabilities. Nevertheless, several research groups have reported remarkable findings using in situ radiography using laboratory sources to capture a wide range of solidification processes (e.g., [21,35,38,52,53,54,55,56], see detailed examples in Section 3.2 below). 

For a synchrotron ISO experimental unit, the design, planning, and preparation stages of the in situ experiment are crucial to the success of allocated beamtime, and getting quantifiable 4D data sets is extremely challenging. The ISO units should be portable, easily assembled, and remotely controlled; ideally via integration into the synchrotron beamline’s control system. Likewise, the specimen environmental chamber is designed to facilitate the easy replacement of samples. The thermal and mechanical systems are required to be stable and vibration-free to minimise motion artefacts in the acquired images. It is always desired to execute several ex situ experiments apriori in the laboratories, including the microscopic examination. In addition, the reproducibility of cell motion, temperatures, atmosphere control, etc. is critical to obtain consistent data. The displacements are often expected to have an accuracy greater than the resolution of the imaging technique (often sub-micron). The sample holding materials are designed to have low X-ray attenuation, and to prevent heat loss. Due consideration to prevent interference from inactive components of the cell are usually ensured, with suitable arrangements for capturing background noise (flat field) in the absence of the specimen. 

Figure 2 shows a schematic of a typical experimental environment of an ISO cell. Given that each ISO cell has different processing capabilities, they will be selected to match the cooling rates and thermal gradient of the solidfication process. Common heating techniques use lasers, infra-red lamps, and resistive/RF heating units to induce heating and melting of metallic samples (see examples in [40,57,58] and later sections). Laser-based sources provide direct, localised heating on the surface of the crucible holding the sample, and enable fast heating (up to 100 °C/min) and cooling rates (e.g., [25,58]). A number of studies have predominantly used resistive [59] and infrared [37] heating chambers enclosed around the specimen (as shown in the schematic of Figure 2). Resistive furnaces are relatively inexpensive, but heating rates are limited to a few degrees (°C) per second, while infrared furnaces can achieve faster heating rates (on small samples) but are less stable and expensive. Similarly, mechanical rigs are needed to perform various modes of deformation such as tension, compression, and torsion/shear, or a combination of these (e.g., [37,60,61,62] and Figure 2b). Several studies have reported integrated electromagnetic, and ultrasound heating and mixing systems (e.g., [63] and Figure 2c). The choice of deformation speed is often limited by the beamline’s imaging hardware. Currently, these ISO mechanical stages can perform deformation at strain rates typically up to 100 µm/s for tomographic capture [64,65]. 

The acquired images are typically processed using image analysis algorithms, to remove noise, artefacts, and to segment features for quantification. Artefacts normally appear in the form of rings [66], centring errors and beam-hardening. Image processing steps such as registration, filtering, segmentation, and quantification, are performed using a number of tools and software applications (refer to [13,22,27,32,61,67,68] for more examples). Some of the recent quantification methods combined with machine learning-based trainable algorithms for feature recognition (such as weka segmentation (e.g., [29,69]), and Seo et al., [70] are proven to be extremely helpful in working with noisy data to identify features on large data-sets ([71]). A single in situ tomographic experiment often results in few tens of 3D scans, and thus, robust image processing algorithms are crucial for processing a large number of data sets collected over several experimental cases. 

## 3. X-ray Imaging of Solidification Processes

In the following section, a brief history of the evolution of X-ray imaging for in situ observation of solidifying metals and alloys is presented. This is followed by a review of recent work on solidification and semi-solid deformation studies. As the number of research articles on this topic is quite large, we focus on those that have either introduced a new ISO technique or a new physical mechanism to the best possible extent. 

### 3.1. X-ray Imaging of Solidification Microstructures

The conventional metallurgical inspection of cast components enabled extensive measurements of final primary and secondary dendrite arm spacing, average grain size estimates and identification of sizes and shapes of defects such as porosity, hot-tears and segregation. However, the transients of evolution kinetics of these features are critical to understand mechanisms and make theoretical predictions, so that the processing is optimized for the best possible component properties. These two are the key motivating factors behind performing real-time observations of solidification processes.

#### 3.1.1. Imaging Using Laboratory X-ray Sources

Following pioneering works of Hunt and co-workers [72], who performed real-time visualisation and investigation of solidification with the help of transparent analogues, the first-ever observation of planar solidification in metallic Al-Au systems was reported by Forsten and Miekk-oja in 1967 [73]. This was the first to report the evolution of a solid-liquid interface, as well as providing an unprecedented experimental validation of the theory of constitutional undercooling, that opened up prospects of theoretical verification as well as generating direct validation cases for numerical tools. Using transmission electron microscopy (TEM), Glicksman and Vold directly observed eutectic solidification and melting at triple point junctions [74]. In the early 1970s, studies by Miller and Beech reported a high-resolution visualisation of equiaxed dendrites in 300 µm thick Al-30 wt% Cu specimens [16] (Figure 3a). Kaukler and Rosenberger [75] performed experiments on Al alloys and reported the evolution of interface morphologies, solute accumulation, and formation of droplets, with a spatial resolution of 70 µm. 

Several researchers subsequently reported two-dimensional observations of dendritic solidification in model alloys of aluminium, for example, Stephenson and Beech [76], Lee & Hunt [77], and Curreri & Kaukler [78]. These studies reinforced the dendritic tip growth characteristics which were otherwise validated with transparent analogues having large Stefan numbers, and extremely low thermal diffusivity. The theoretical relationships between the dendrite tip radius and the velocity (marginal stability) and the microscopic solvability were needed to be demonstrated with metallic in situ experiments [1]. Further, the mechanisms of the formation of defects (such as porosity and segregate channels) during alloy solidification had not been understood. For example, Chalmers [79] and Tiwari and Beech [80] reported the occurrence of hydrogen porosity in aluminium castings in different sizes and shapes, however the growth mechanisms were only speculated. Giamei and Kear [81] reported presence of freckle channels in superalloy ingots, however, direct observations of the microstructural origin and the role of cooling rates were not possible without ISO experimentation. Lee and Hunt [77] reported the first observations of hydrogen porosity in Al alloys, visualised through a custom-built X-ray Temperature Gradient Stage (XTGS) (Figure 3b). The authors also highlighted the role of solidification microstructures in controlling the pore morphology. The observations were used not only to develop physics-based predictive tools, but also used to validate their microstructure and pore growth models [82]. These models were further developed to predict microstructural evolution in as-cast and weld components, including the prediction of hydrogen porosity defects [83,84,85,86]. It is worthwhile mentioning that laboratory X-ray imaging of solidification processes continues to be one of the key experimental approaches to develop new understanding and explore the solidification behaviour. For example, some of the recent laboratory-based radiographic [68,87,88] and optical [89] visualisation has been extremely insightful, resolving not only microstructural features but also associated flow phenomena, such as freckle channels [17,87] (Figure 3c), including high-density metallic alloy systems, validating hypotheses [90,91] and 3D models.

**Figure 3 materials-14-02374-f003:**
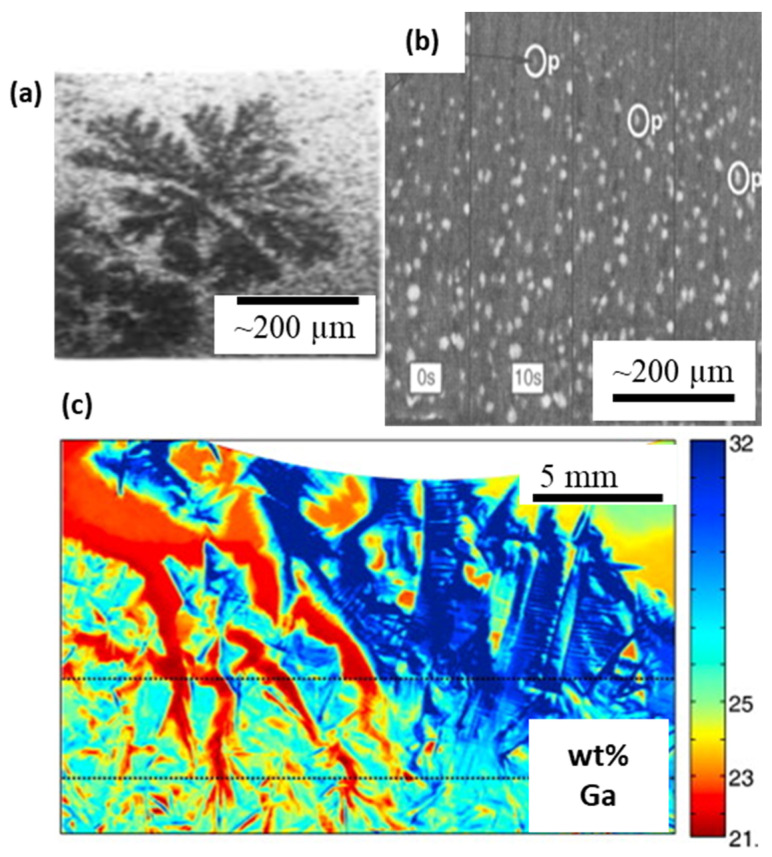
In situ radiography experiments in laboratory X-ray sources, (**a**) equiaxed dendritic solidification of Al-30wt%Cu alloy (after Miller and Beech [16]). (**b**) evolution of microstructures and hydrogen porosity during solidification of Al-Cu alloy (after Lee and Hunt [77]). (**c**) Real-time observations of freckle formation in Ga-In alloys, (after Boden et al. [87]). Permitted for reuse from respective publishers.

#### 3.1.2. Synchrotron X-ray Investigations of Solidification

The first synchrotron observation of dendritic solidification was reported by Matsumia et al. [92], who observed the solidification of silicon steels. The authors employed a custom-built solidification cell in Japan (KEK, PF) and observed evolving solidifying interfaces. Further, Billia and co-workers [93] were able to resolve the scales of secondary arms in Al-Cu alloy (Figure 4a), and paved the way for a whole new approach to investigate real-time solidification behaviour of metals. They reported topographic observations of dendritic growth in Al-Cu alloy, with 40 × 6 mm^2^ FOV, 200 µm thickness, and exposure time of about 2 s. With beamline energy up to 12 keV, a spatial resolution of a few micrometres (Figure 4a). These were further improved by Arnberg and co-workers [94] at the same ESRF beamline, and showed time evolution of solute distribution and dendritic solidification (Figure 4b). Wang et al. [95] reported in situ observations of Fe intermetallic growth for the first time, using synchrotron radiographic imaging at SRS Daresbury Laboratory in the UK. They used white, parallel beam X-rays at 34 keV to image a 1 mm thick Al-Si-Fe alloy specimen at a spatial resolution of 1 µm and an acquisition frequency of 2 Hz. They employed a novel infrared heater to heat the sample to its liquid state and cool at 20 °C/min. 

With continual upgrades to synchrotron, imaging, and image analysis technologies, the new millennium witnessed a paradigm shift in utilisation of X-ray imaging for in situ solidification studies. The ever-expanding capabilities of X-ray sources in terms of beamline optics, energy stability, and X-ray flux, enabled imaging studies on materials having high-density and higher melting temperatures in extreme and demanding environments. This also enabled mimicking a complete metallurgical process, from casting to advanced thermo-mechanical processing [1] and laser processes, such as welding (e.g., [8]) and AM (e.g., [97]).

Maire et al. [40] demonstrated several examples of high-resolution X-ray synchrotron tomography for metallic systems, which led to a large number of studies on microstructure evolution during solidification. The authors showcased ESRF-ID19′s capabilities as a microtomography beamline for capturing microstructural growth in Al alloys. Spatial and temporal resolutions of about 1 µm and 0.3 s were achieved in performing high-resolution tomographic scans within an hour (Figure 5a). Matheisen and co-workers reported many new and interesting phenomena by using radiographic observations of solidification of a range of Al-Cu alloys, specifically capturing dendrite fragmentation, overgrowth, interdendritic porosity, and liquid-phase separation in monotectic reaction [53,96,98]. Similar directional solidification experiments were reported by Yasuda and co-workers in Sn-Bi alloys [99,100], and Nguyen Thi and co-workers also published a number of solidification mechanisms in Al alloys [24]. 

The 3D geometrical quantification of dendritic microstructures, particularly the curvatures of the tip radii, the orientation selection etc., needed to be dynamically quantified for understanding the kinetics of growth and coarsening. Fast tomographic capture of metallic materials in situ was made possible with synchrotrons upgrading to 3rd generation, combined with significant improvements in beamline hardware, detectors, image acquisition, and reconstruction methods. A detailed study of capabilities of 3D X-ray microtomography for metals processing was reported by Salvo et al. [101], demonstrating in situ evolution of solidification microstructures and metal foams (Figure 5a–c). They used a continuous acquisition of tomographic scans in ID19 beamline at ESRF, in intervals of about 30 s, collecting a total of 50 scans. The specimen was heated to 660 °C, and cooled at approximately 3 °C/min until the solidus temperature of 545 °C was reached. A key revelation from the study was the understanding of actual dependence on the mushy zone permeability during the evolution of the semi-solid, suggesting key improvements over the conventionally used Carmen-Kozeny model. Fife and co-workers [21] reported a new laser-based in situ solidification kit, integrated with the TOMCAT beamline at the Swiss Light Source (PSI-SLS). These authors used a polychromatic X-ray and a high-speed pco. Dimax camera that allowed an exposure time of 1–5 ms, and 1–3 μm pixel resolution. Figure 5d–f show the reconstructed and segmented volumes of dendritic microstructures during a melting and solidification experiment. Thus, a tomographic scan could be captured in less than a second, which opened up a range of new experimental possibilities. The precise heating by the lasers also allowed enhanced cooling rates by about an order of magnitude (30 °C/min). 

In recent years, there have been several key observations of microstructural evolution that have helped develop detailed geometrical [102] as well as a physical understanding of dendritic [103] and eutectic [31,104] growth morphologies. Matheissen and co-workers investigated the nucleation and growth of Sr-modified irregular eutectic microstructures in Al-Si-Cu-Sr alloys [31] (Figure 6a). The radiographic observations helped reveal the mechanisms by which intermetallics and eutectics nucleate and grow between primary dendrites. Puncreobutr et al. [33], investigated the nucleation and growth and of Fe-intermetallics in Al alloys using fast tomography (Figure 6b,c). The authors were able to resolve 10 μm thick intermetallic features, and investigate their role in altering the mushy zone permeability and the associated hot-cracking during solidification, producing an enhanced model of permeability. 

Despite several insightful observations of solidification using aluminium-based model alloy systems, studies of high temperature and reactive metals continue to be extremely challenging. Only a few groups have developed capabilities to perform imaging of high-temperature materials (typically above 1000 °C), including processing of ceramics. Such studies on high-temperature materials were required to establish process-microstructure relationships for materials such as steels, cast iron, superalloys and ceramics. The mechanisms of peritectic transformation and graphite nodule growth of classical iron-based alloys were only indirect and speculative until real-time experiments demonstrated the phenomena in real-time. Yasuda and co-workers developed radiographic and tomographic experiments of steels, shed insights into the formation of peritectic transformation, shear deformation, and solidification of low carbon steels [35,62,100,105,106] (Figure 7a). The authors used the imaging beamline at Spring-8, integrated with novel high-temperature rigs, that enabled fast radiographic and tomographic capture in controlled environments with a resolution of 5 μm/pixel for a 100 μm thick specimen. Azeem et al. [23] first reported 3D imaging of microstructural evolution during the solidification of high-temperature Fe, Co, & Ni-based alloy systems, with hafnium as a high X-ray attenuation model alloying addition. Specimens of 1 mm diameter in size were encapsulated in inert-gas filled capsules and were imaged at the I12-JEEP beamline at DLS, with a spatial resolution of 1.3 µm/pixel and an exposure time of 0.1 s, leading to a scan time of the order of 40 s (Figure 7b). A similar set-up was also used to observe the formation of spheroidal graphite nodules in cast iron, capturing the dynamics of the evolution of nodules for the first time [107]. The combination of high resolution and fast acquisition, coupled with improved analysis methods have also evolved to identify geometrical characteristics of evolving features in metals, such as curvatures of dendritic tips, and other phases [102,108]. Recently, Reinhart et al., [59] demonstrated the imaging of Ni-based alloys during solidification, clearly identifying columnar directionally solidified microstructures and solute channel formation (Figure 7c). The authors used a CMSX-4 superalloy specimen in a gradient furnace to capture real-time observations at about 0.1 s time intervals.

### 3.2. X-ray Imaging of Semi-Solid Deformation

The formation of several casting defects are linked to the deformation of the mushy region [109,110] and are integral to the studies of solidification processes. The role of microstructures in initiating such defects, particularly the role of shear banding and dilation of the inter-granular spacing was not understood. Therefore, semi-solid deformation of partially molten alloys has been investigated by several researchers to explore the following key aspects: (i) microstructural response to deformation (e.g., [111]); (ii) granular behaviour of the semi-solid mixture, including Reynold’s dilatancy (e.g., Gourlay and Dahle [34]); (iii) deformation of the microstructures such as equiaxed [61], columnar [112], and globular [22,113] morphologies; and (iv) the role of deformation on defects such as hot-tearing or hot-cracking (e.g., [52,114]), intermetallics [33] and porosity [29]. 

Maire et al. [14,40], integrated a tensile rig with the microtomography beamline and showed capabilities in capturing the microstructural response to material deformation in Al alloys. Spatial and temporal resolutions of about 1 µm and 0.3 s, respectively, were achieved via performing high-resolution tomographic scans. A comprehensive literature review of imaging-based studies on hot-cracking of light alloys can be found in [115]. Phillion et al. [116] precisely measured the temperature at a hot-tear, and motion of liquid into the tear using in situ radiographic imaging, using a custom-built IR heater and a bespoke loading rig. In 2009, Terzi et al. [117] reported first-ever in situ tomographic capture of deforming semi-solid Al-Cu alloy (Figure 8a–c), that followed a prior ex situ microtomographic investigation of hot-tearing by Phillion et al. [52,118]. Terzi et al. combined a heating stage with a tensile rig in a beamline at ESRF, performing isothermal tensile deformation of a semi-solid metallic specimen. A spatial resolution of 2.8 µm was achieved while taking tomographic scans of 400 radiographs in about 27 s. They reported real-time capture of a hot-tear defect appearing in specimens having 25% liquid, with displacement speeds of 0.1 mm s^−1^. The occurrence of dilatant bands and other deformation mechanisms in equiaxed structures was observed in situ by Gourlay et al., [119] (Figure 8d,e). A similar study also demonstrated solidification and semi-solid deformation of Al-based alloys, revealing faster 3D capture of deformation-induced defect formation [55]. 

Lee and co-workers developed a custom-built tension-compression-torsion rig (termed the P2R), and a resistance heating furnace (Etna), and demonstrated capabilities in capturing 3D plus time evolution of damage during solidification and semi-solid deformation [32,120]. The P2R rig enables precise deformation control in tension, compression and torsion (1–500 N, 1–10^4^ µm/s deformation speeds) whilst rotating with better than 50 nm concentricity using air bearings, enabling 1 μm voxel tomographs. The Etna furnace enables temperatures up to 1000 °C, with heating/cooling rates of approximately 1 °C/s, while a separate furnace, Alice, reaches 1600 °C [23,65,107]. A further study with similar set-up also enabled the investigation of intermetallics in 3D plus time, showing precise locations and conditions of intermetallic nucleation and growth [121]. Another study investigated material flow, deformation, and strain imposed on the microstructure using the P2R and furnace, together with digital volume correlation (DVC) tools to shed important insights into microstructural behaviour of metals [29]. Similarly, recent deformation studies have shown how deforming equiaxed semi-solid alloys can cause dilatancy and increase hydrogen porosity formation due to combined diffusion and pressure conditions [29]. Such approaches have found key mechanisms of defect formation, such as shear banding and porosity, thereby helping to improve processing methods from conventional sand cast to high-pressure die cast (HPDC) components.

A number of studies have created an extensive database of semi-solid deformation mechanisms, particularly dilatant shear bands [34,106,122,123,124], granular motion [61,111,113,125,126,127], and associated new observations at the microstructural level [22,37,41,128]. These were achieved with the help of various in situ deformation techniques such as compression [61,113], pure shear [34], extrusion [111,126] and indentation [22,129,130]. In ref [35,62], authors were able to successfully capture the deformation of carbon steels in the semi-solid state, using a specially designed ISO cell for steels, integrated with the Spring-8′s imaging beamline. Furthermore, such experimental approaches have shown new micromechanics in several metallic and non-metallic systems. For example, Bale et al. [37] performed real-time quantitative imaging of failure events in materials under loaded conditions, and at temperatures of 1600 °C. Using a high-temperature and loading environment integrated with the beamline, the authors reported real-time capture of cracking in high-temperature ceramics. Similarly, using laser-based heating and fast tomographic capture at SLS, self-healing of cracks in advanced ceramics was reported [131].

One of the significant contributions of in situ imaging to date is in transforming the grain nucleation and growth models, by providing new insights and validation cases. The hardware and ISO instrumentation capabilities can capture additional aspects, including the multi-physics, multi-modal, and multi-scale imaging of materials, spanning a large number of applications in the solidification processing of metals. Fast imaging capabilities have been able to shed tremendous insights into the clustering of nanoparticles, cavitation in metals, and grain fragmentation under the influence of external forcing such as electromagnetic, ultrasonic, and other chemical reactions. The role of electromagnetic forcing on the fragmentation of dendrites was reported by Liotti et al. [63] (Figure 9a,b), and with laboratory X-ray sources by Eckert and co-workers [88,132,133,134]. Similarly, a number of studies have investigated the influence of ultrasonic processing on liquid metals and grain growth [39,135,136,137,138]. Wang et al. [139] showed how an oscillating cavitation bubble interacted with the solidifying interface in Bi-Zn alloy, captured the phenomena that occur in the order of milliseconds (Figure 9c,d). Other similar studies have also shown synchrotron studies of particle agglomeration and clustering, as well as the removal of clusters using ultrasound cavitation. 

X-ray tomography has been used for a large number of applications to understand how zinc and lithium batteries degrade [9,140,141] (Figure 9e). Deville et al. reported fast tomographic capture of nucleation in colloidal suspensions [142]. Liotti et al., [143] analysed a series of radiographic images to develop machine-learning driven models of nucleation in metals. A handful of studies have reported on the uses of combined X-ray imaging and diffraction to investigate the sequence of nucleating phases, as well as to obtain the data for feeding thermodynamic databases and models [12,144]. Coupled diffraction imaging studies have also been reported to understand the onset of new nucleating phases during solidification [13,138,144,145]. The nucleation and growth of primary phases and the role of nucleating TiB2 particles were investigated by Iqbal et al., [146] using in situ X-ray diffraction (Figure 10a,c). The authors used a monochromatic X-ray beam with an energy of 70 keV (wavelength of 0.177 Å) and a beam size of 200 × 200 μm^2^, with the sample covering 5 mm diameter of the sample (with a height of 10 mm) that was mounted in a glassy carbon container within a vacuum furnace. Narumi et al., [145] reported combined tomographic microstructural capture and diffraction-based crystallographic information using a novel coupled technique, demonstrated using tensile deformation of Al-Cu alloy (Figure 10d–f).

## 4. The Future? In Situ Imaging for Ultra-Fast Solidification Processing, Additive Manufacturing

An emerging application of X-ray imaging is in understanding the complex, multi-phase and multi-scale AM processes. AM has shown tremendous promise in flexible metal processing, and AM components are replacing assemblies in various aerospace (for example, General Electric’s 3D printed aeroengine, [147]) and medical applications (such as implants and devices, [148]). The existence of powders and solidified structures in the solid phase, multi-component liquid, vapour plume (metal vapour and cover gas), and plasma, all occurring in a very short time-span and high energy density process, makes it extremely complex and challenging to study. 

Laser additive manufacturing (LAM), such as laser powder bed fusion (LPBF) and directed energy deposition (DED), is an emerging digital manufacturing technology that builds up 3D components using powder feedstock and a focus laser beam, layer-by-layer, directly from a digital file, e.g., CAD and stl files, etc. In contrast to traditional manufacturing technologies, such as casting and subtractive technologies, LAM components can be made with a high degree of design freedom, mass customisation, intricate features, and short lead-times. This makes LAM attractive to a wide range of interests from aerospace, automotive, biomedical, defence, and energy sectors [6,149,150]. 

The uptake of LAM technologies for the production of safety-critical applications, e.g., blisk turbine engine components, is currently hindered by many technical challenges, including undesired microstructure (e.g., porosity and hot cracking), delamination, and detrimental residual stresses in LPBF parts, some of which lead to poor mechanical performance and potential build failure. The microstructural evolution is governed by the laser-matter interaction which occurs in an extremely short timescale (<100 ms). During this period, powder materials undergo sintering, melting, mixing, reactions, vaporisation, and rapid solidification, and hence the fundamental understanding of the LPBF remains not well-understood. The heating/cooling rates of the LPBF process are approximately 10^5^–10^6^ K s^−1^ which is three orders of magnitude faster than that of traditional casting, and the thermal gradient established inside the molten pool can reach between 10^3^ and 10^4^ K mm^−1^. To tackle such demanding scientific challenges, research teams across the globe developed different types of additive manufacturing simulators [13,26,97] combined with high-speed X-ray imaging and diffraction facilities at Advanced Photon Source (APS) [151], Diamond Light Source [26,58,152], European Synchrotron Radiation Facility [153], Stanford Light Source [13,154,155], and Swiss Light Source [42], to probe and elucidate the molten pool dynamics during LPBF. 

### 4.1. In Situ and Operando X-ray Imaging of LPBF

Cang et al. [156] simulated the onset of the laser-mater interaction to study melt pool geometry, solidification rate, and phase transformation of Ti-6Al-4V during single spot LPBF events (ca. 1 ms per event). They revealed that the formation of keyhole pores is due to insufficient liquid feeding in the melt pool. Cunningham et al. [8] further studied the development of vapor depression and keyhole formation during LPBF of Ti-6Al-4V under various laser power densities and processing regimes (from the keyhole to transition and conduction modes). The evolution of the vapor depression is summarised in the following stages: (i) melting, (ii) vapor depression formation and growth, (iii) vapor depression instability, (iv) keyhole formation and growth, and (v) melt pool shape change. Other groups have also employed high-speed X-ray imaging to study the melt pool geometry as a function of input energy densities or build parameters [27,157], varying the pressure of the cover gas [148], and oxygen species in the powder composition [152]. 

In addition to the melt pool geometry studies, several recent studies focused on the fundamental origin of the keyhole dynamics and the evolution of keyhole porosity during LPBF in a single layer build [158,159] and multi-layer build conditions [153,160]. Cang et al. [159] suggested that the keyhole porosity is increasingly sensitive to scan speed. They also suggested that the keyhole porosity regime varies slightly, which is also supported by a high fidelity simulation [158]. In contrast, Sinclair et al. [160] revealed that the powder layer thickness alters the track height in LPBF, leading to inconsistent laser melting in subsequent build layers. They also revealed that keyhole pores can be removed via laser remelting. However, an inadequate laser penetration depth would lead to insufficient liquid feeding to fill pre-existing pores, forming irregular pores inside the additive manufactured samples.

Besides the keyhole pore formation, there are many different types of porosity exhibited in LAM components, including gas pores, open-pores, lack of fusion, inter/intralayer pores, etc., and several research teams investigated their underlying formation mechanism, see examples in Figure 11. Leung et al. revealed the melt pool and melt track dynamics inside single-layer tracks (with a track length of 5 mm) and subsequent build layer during LPBF of Fe-based alloys (Invar 36 with [152]/without powder oxidation [58] and SS316L [26]) under overhang conditions. They uncovered a series of pore evolution mechanisms during LPBF, including pore formation due to the keyhole mode operation, a reduction of gas solubility in the molten pool (Figure 11b), pore migration via the recirculating Marangoni-driven flow, open-pore formation by pore bursting (Figure 11c), pore dissolution and dispersion by laser re-melting in subsequent build layer [58]. They also showed that pores can also be formed via vaporisation of low boiling point elements i.e., the reboil effect [26] and heterogeneously nucleate from the metal oxides’ surface inside the melt pool [152], and pore growth via coalescence of smaller pores. Aiden et al. [154] reported that pores can also be formed during laser turning due to the collapse of a keyhole depression or caused by abnormal heat input to the powder bed due to the acceleration/deceleration of scanning mirrors (Figure 11d). Hojjatzadeh et al. [25] captured three other pore evolution mechanisms, including (1) pore transfer from powder feedstock to the molten pool, pore trapped by (2) fluctuations of the melt surface or at the depression zone (Figure 11e,f), or (3) pore formation from a crack (Figure 11g).

Combined with X-ray imaging and object tracking techniques, the average melt pool velocities were first estimated to be in the range of 0.1–0.4 m s^−1^ under overhang conditions [58] and 0.6 m s^−1^ during laser remelting [161]. Hojjatzadeh et al. [25] and Quo et al. [162] performed detailed studies on the flow pattern of an entire melt pool using tracer particles under various industrial conditions. They divided the melt pool into (1) laser-interaction, (2) transition, and (3) circulation domains. Notably, their reported average melt flow velocity of the entire melt pool was 1.1 ± 0.5 m s^−1^ and very similar to prior work [58,161,163]. The aforementioned studies showed that most flow patterns followed a centrifugal Marangoni convection owing to the negative temperature dependant coefficient of surface tension in metallic systems [58,162,163,164,165]; however, the Marangoni convection can be reversed when there is an increase in oxygen content within the molten pool, i.e., centripetal Marangoni convection [152], resulting in a deeper molten pool, larger pore size distribution, and more frequent spatter ejection. These new insights provided detailed information regarding the influence of powder chemistry on the melt flow and defect dynamics under a wide range of processing regimes which can be used for developing a reliable high-fidelity simulation model to predict the formation of undesirable features during LAM.

By studying the fundamental mechanisms of pore formation, several mitigation strategies have been proposed to reduce or eliminate porosity in LPBF components: (1) using laser re-melting techniques to reduce pore size distribution and number density either by promoting gas release from the keyhole or by inducing liquid metal flow to partially or completely filling pre-existing pores [152,160]; (2) reduce the input normalised enthalpy to reduce metal vaporisation during laser turning [154]; (3) use the Marangoni-driven force to drive porosity towards the laser-interaction domain (with a wide melt pool); and (4) reduce the melt viscosity (10^−3^ Pa S) to avoid pore trapping [26] by maintaining the melt temperature just below the boiling point.

### 4.2. In Situ and Operando X-ray Diffraction of LPBF

In situ and operando X-ray diffraction or X-ray imaging in the reciprocal spacing has been used to capture the dynamic phase transformation, extract the subsurface cooling rates, and evolution of strains during LPBF of Ti-6Al-4V [42,155,156]. Cang et al. [156] studied the phase transformation during a single spot LPBF of Ti-6Al-4V, showing the evolution powder underwent melting, solidification (with a cooling rate of 1 × 10^5^ K s^−1^), and reported that β → α’ solid-state phase transformation is diffusionless and has a transformation rate above 10^4^ μm s^−1^. Thampy et al. [155] studied the effect of laser powers on the changes in lattice spacing during LPBF of a single layer Ti-6Al-4V track. They reported that the lattice parameter and residual strain of the β-Ti reduced with increasing cooling rate from 1.5 × 10^4^ to 7.5 × 10^4^ K s^−1^, similar to that reported ref [42] whereas the α/α’ phase is invariant with cooling rate. To capture the full complexity of the layer-by-layer laser melting process, Hocine et al. [42] recorded diffraction patterns during LPBF of Ti-6Al-4V rectangular layer, in which each layer consisted of 33 scan vectors (with a length of 2–8 mm) with a hatch distance of 60 μm. They found that reduced scan vector length allowed β-Ti to remain longer in the melt pool, accommodate more strain, and reduce the overall residual stress of the part, in agreement with ref [155]. They also emphasised that the changes in lattice spacing of β-Ti were solely due to thermal contraction rather than by phase transformation. Undoubtedly, these studies provide crucial information to advance some aspects of microstructural simulations for the predictions of phase precipitations, phase transformation, and residual stress development. However, it remains very challenging to deconvolute and interpret the thermal, chemical, and stress effects from these dynamic diffraction patterns.

## 5. The Outlook of In Situ and Operando X-ray Imaging

ISO X-ray imaging has been used as a major workhorse to investigate a wide range of solidification processes, including casting, semi-solid deformation, and laser material processing, provide mechanistic information to test prior hypotheses and also advance new understanding of complex solidification phenomena. This also means that they will allow us to capture rare events occurring in physical systems, including nucleation and growth, solid-state transformation, and precipitation events. The synchrotron facilities and techniques are undergoing massive upgrades that will enable imaging of large components, including human body parts, at unprecedented resolutions and levels. For example, ESRF has rebuilt the brightest-to-date X-ray source, ([166]), and the same facility has demonstrated new hierarchical phase-contrast imaging techniques ([167]). Such advances will enable scanning of a full aero or automobile engine or replicate large-scale phenomena in an imaging beamline. There is also a growing trend to combine X-ray imaging (in real-space and reciprocal space) with ultra-fast optical, thermography, and Schlieren Imaging, i.e., multi-modal imaging, to study a wide range of interactions between solid, liquid, and vapour phases during solidification. Further, nanoscale phenomena such as nucleation kinetics and orientation selection can be investigated only when required spatial and temporal resolutions are achieved. With recent advances in X-ray imaging instruments, such as upgrades of synchrotron radiation facilities and X-ray Free-Electron Laser (XFEL) facilities, these new imaging technologies will enable studies of structural dynamics at a time scale down to femtoseconds and at the nanoscale, covering electronic dynamics, lattice dynamics, and the formation and breaking of chemical bonds. In summary, the outlook of X-ray imaging will remain as a core advanced characterisation technique to unravel discoveries into matter and dynamics with impact across a wide range of scientific fields.

## 6. Concluding Remarks

This paper reviews recent developments in the field of in situ and operando techniques for real-time 2D/3D/4D X-ray imaging of solidification processing of materials. The ISO experimentation has been primarily motivated by the quest for capturing real-time dynamics of evolution of microstructures and defects, and obtaining validation for theoretical and numerical predictions. An overview of several key approaches and resulting scientific revelations are presented. The studies highlight the advancement in the form of acquiring complete 3D scans in a few seconds and the integration of complex instrumentation for in situ and operando imaging in synchrotron beamlines. A brief overview of emerging technologies such as additive manufacturing, and outlook of imaging as an advanced characterisation tool is presented. This review not only provides a critical understanding of ISO experimentation to capture challenging phenomena, but also presents a detailed overview of the real-time observations of materials processing studies.

The outlook for such experimental research appears stronger than ever due to the following factors; (i) newer approaches that combine multiple techniques such as simultaneous imaging and diffraction, (ii) multi-scale imaging techniques using hierarchical imaging and (iii) operando cells simulating extreme environments in beamlines with continuous upgradation in the x-ray source and imaging hardware. Further, the acquisition as well as analysis of the imaging data is being made easier by combining data-driven and iterative machine learning approaches.

## Figures and Tables

**Figure 1 materials-14-02374-f001:**
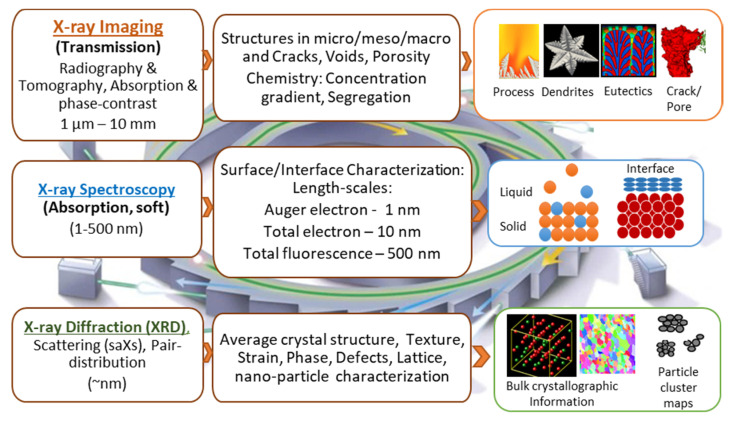
Some of the important synchrotron X-ray based, in situ, and operando characterisation techniques for metals, their scales, capabilities, and examples (adapted from Bak et al., [9]).

**Figure 2 materials-14-02374-f002:**
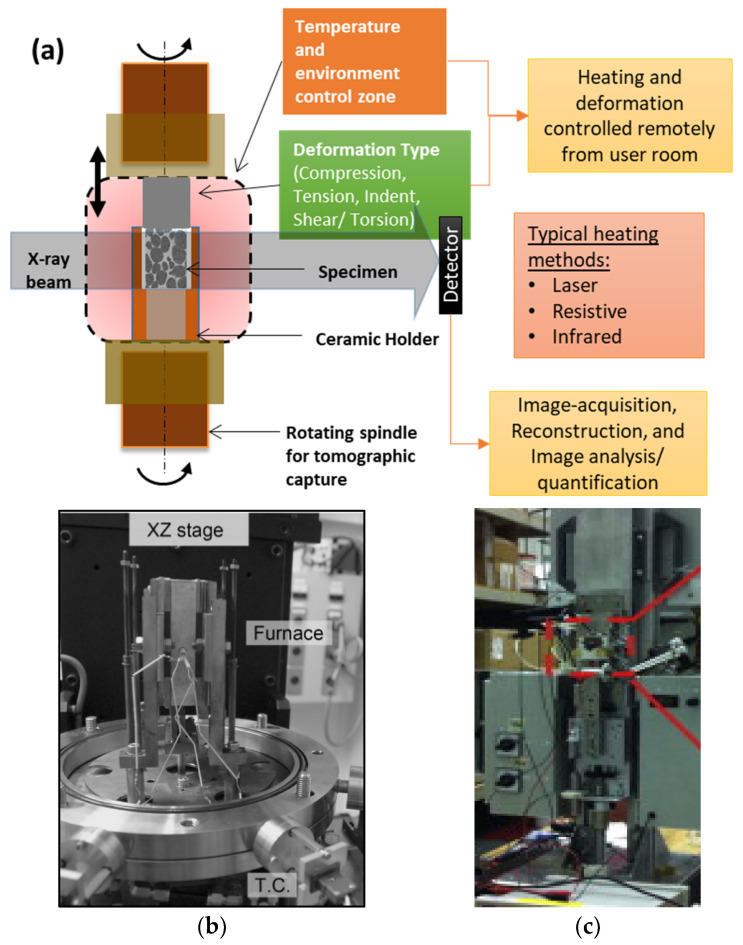
Examples of in situ experimental set-up for X-ray imaging: (**a**) schematic of a typical thermal-mechanical stage for tomographic imaging, (**b**,**c**) Photographs of experimental set-ups from some of the research groups (**b**) after Yasuda et al., [62] in Spring-8 (Reproduced from Yasuda et al., Development of X-ray Imaging for Observing Solidification of Carbon Steels, ISIJ International, 2011, 51:402–408. Copyright © 2021, The Iron and Steel Institute of Japan, All rights reserved.) and (**c**) after Liotti et al., [63], at DLS. Permitted for reuse from respective publishers.

**Figure 4 materials-14-02374-f004:**
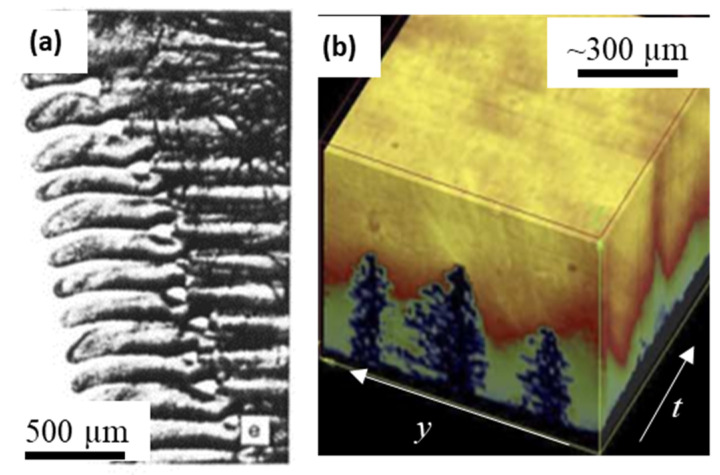
Early in situ synchrotron experiments of solidification: (**a**) white-beam X-ray topographic view showing melt growth of Al-Cu alloy (after Grange et al., [93]); (**b**) Mushy zone and liquid volume contours (y,z,t) showing compositional distribution during solidification of Al-30 wt%Cu alloy (after Mathiessen et al., [96]). Permitted for reuse from respective publishers.

**Figure 5 materials-14-02374-f005:**
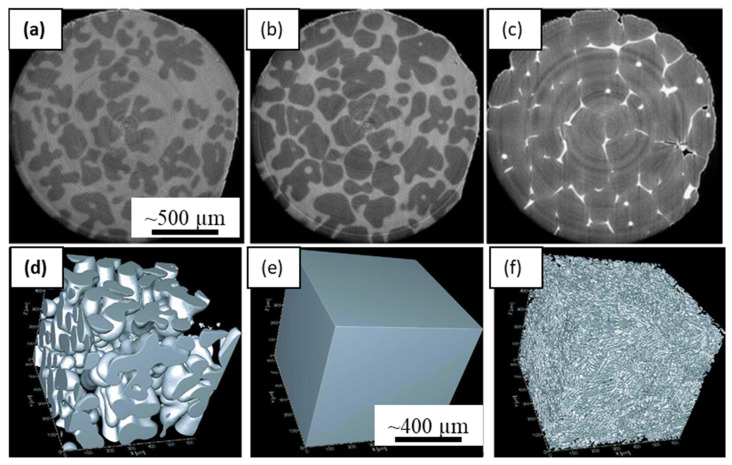
In situ tomographic capture solidifying microstructures: (**a**–**c**) Reconstructed slices of tomographic scans captured during solidification of Al-4wt%Cu alloy, showing microstructure evolution in real-time. The light grey shows liquid and the dark phase represents solidifying microstructures, after Salvo et al., [101] (Reproduced from Salvo et al., 3D imaging in material science: Application of X-ray tomography. Comptes Rendus Phys. 2010, 11, 641–649. Copyright© 2021, Elsevier Masson SAS, All rights reserved); (**d**–**f**) 3D reconstructed views of melting and solidification sequence of Al-20 wt.% Cu alloy performed using a laser-based furnace (after Fife et al., [21]). Permitted for reuse from respective publishers.

**Figure 6 materials-14-02374-f006:**
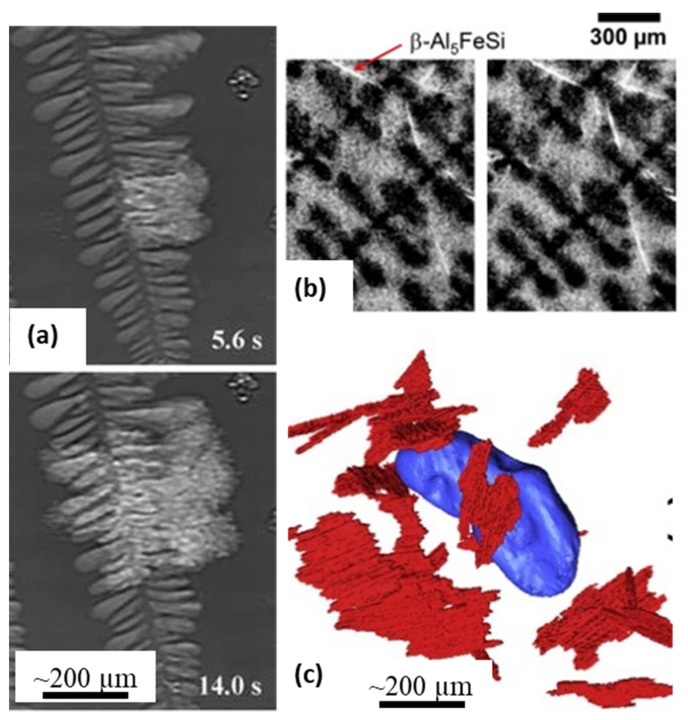
(**a**) Sr-modified irregular eutectic microstructure formation during directional solidification of an Al-Si-Cu-Sr alloy (after Mathiessen et al., [31]) (**b**) 2D slices of the tomographic data showing nucleation and growth of Fe-intermetallics (white) during dendritic (black) solidification of secondary aluminium. (**c**) 3D reconstruction of Fe intermetallics in (**b**) showing the origin and time of nucleation (after Puncreobutr et al. [33]). Permitted for reuse from respective publishers.

**Figure 7 materials-14-02374-f007:**
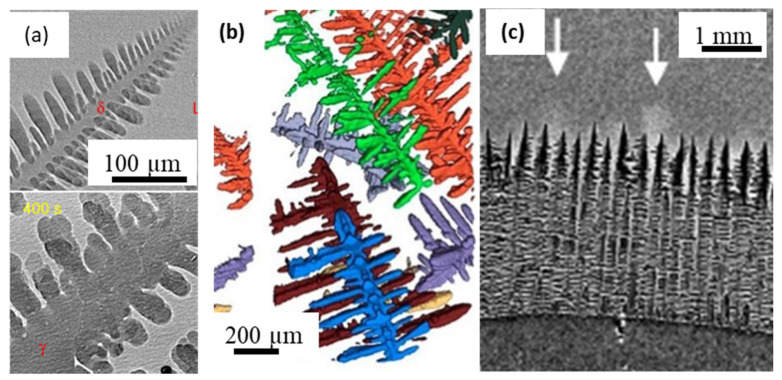
In situ imaging of a high-temperature metallic system: (**a**) Radiographic images showing fragmentation of γ grains induced by a massive-like δ–γ transformation in 0.45 C steel (after Yasuda et al. [105] (**b**) Dendritic pattern formation in Ni, Fe, and Co alloys using fast synchrotron tomography. A cooling rate of 0.1 K/s was employed, with each tomographic scan performed in about 1–2 min, (after Azeem et al., [23]). (**c**) Segregated channels during solidification of Ni-based alloy captured via radiographic imaging, (after Reinhart et al., [59]) Permitted for reuse from respective publishers.

**Figure 8 materials-14-02374-f008:**
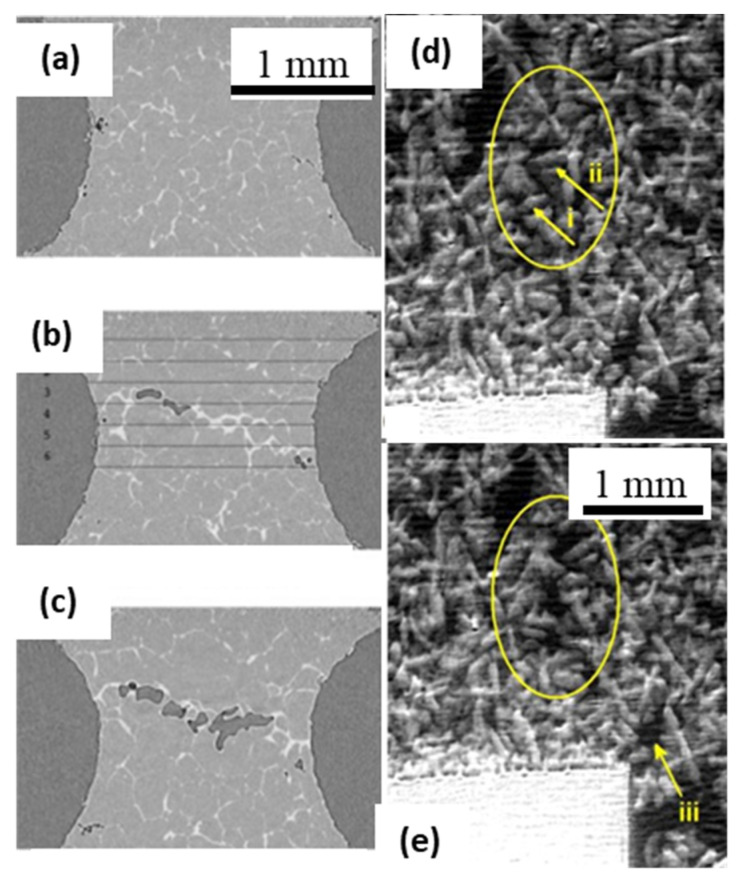
(**a**–**c**) 3D reconstructed slices showing the evolution of a hot-tear during tensile deformation of a semi-solid Al-Cu alloy specimen (after Terzi et al. [117]). (**d**,**e**) In situ radiographic observations showing direct evidence for Reynolds’ dilatancy in an equiaxed dendritic mush. In the circled region, an initially compacted crystal assembly dilates during deformation under a compressive load (after Gourlay et al. [119]) Permitted for reuse from respective publishers.

**Figure 9 materials-14-02374-f009:**
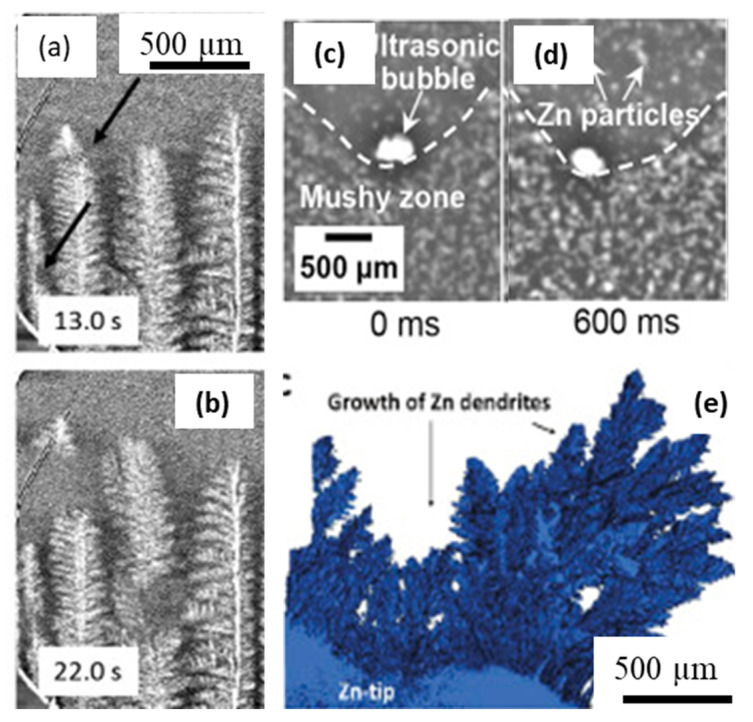
Solidification experiments in the presence of external forcing via electromagnetic and ultrasonic forcing, and electrochemical reactions: (**a**,**b**) solidification sequence of an Al–15 wt.% Cu alloy solidified with an electromagnetic field. Thermal gradient G = 48 K·mm^−1^, peak current I = ±300 mA, pulse frequency 1 Hz, sine-wave form, magnetic field B = 0.1 T, and peak Lorentz force acting on the sample = ±0.3 mN (after Liotti et al., [63]). (**c**,**d**) A sequence of X-ray images, showing the cyclic impact and pounding of the oscillating bubble at the L-S interface in the Bi-8% Zn alloy during ultrasonic cavitation, (after [139] (**e**) Reconstructed 3D cross-sectional image of the dendrites in zinc anode attached to the tip, (after Yufit et al., [140]). Permitted for reuse from respective publishers.

**Figure 10 materials-14-02374-f010:**
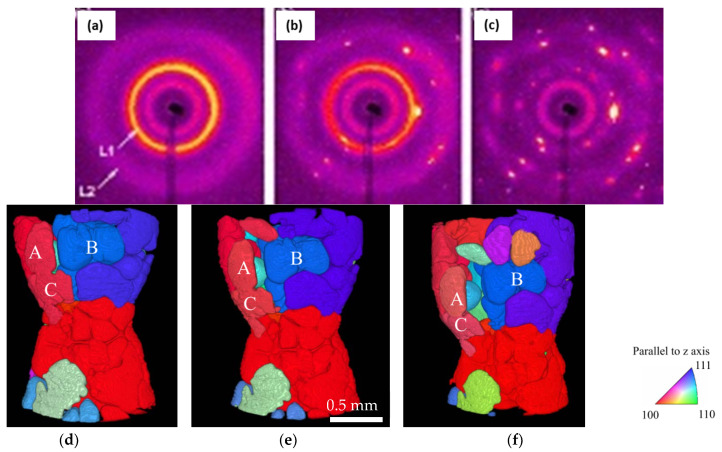
(**a**–**c**) X-ray diffraction patterns of the aluminium alloy with titanium solute (0.1 wt.%) and added TiB_2_ particles (0.1 wt.%) at different stages (**a**–**c**) of the solidification process (after Iqbal et al. [147]). Collected during cooling from 973 K at a rate of 1 K/min; (**d**–**f**) show the crystallographic orientation of the 3D-reconstructed solid grains in the equiaxed Al–10 mass% Cu alloy at the strains of (**d**) −0.04, (**e**) −0.06, and (**f**) −0.13; where the color variation indicates the crystallographic orientation as detected by 3DXRD, after Narumi et al. [146]. Permitted for reuse from respective publishers.

**Figure 11 materials-14-02374-f011:**
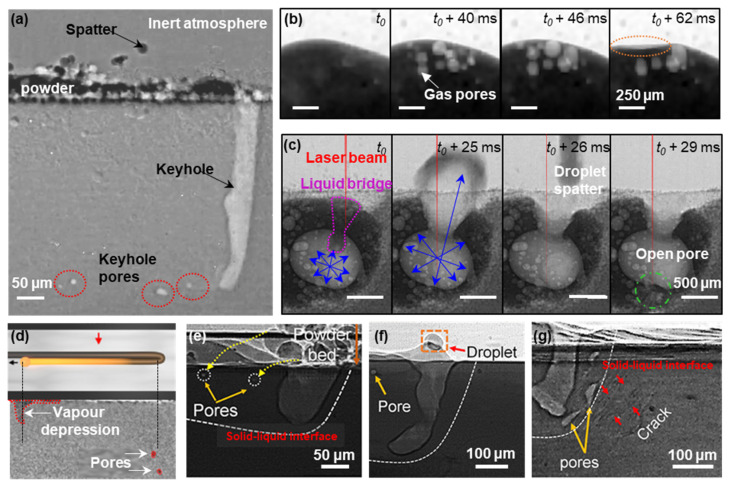
Radiographs captured during LPBF showing different types of pore formation mechanisms: (**a**) formation of keyhole pore due to unstable keyhole; (**b**) formation of gas pores due to reduction in gas solubility as the molten pool cools down (after [58]); (**c**) formation of open-pore due to pore bursting during multilayer laser remelting (after [153]); (**d**) formation of pores due to laser turning and acceleration/deceleration scan mirrors (after [155]); (**e**) pore transfer from powder feedstock to the molten pool (after [25]); (**f**) pore formation due to surface fluctuations (after [25]); and (**g**) pore growth from existing cracks (after [25]). Permitted for reuse from respective publishers.

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
