# Peer review of "Progress on In Situ and Operando X-ray Imaging of Solidification Processes"

_materials, 2021, doi:10.3390/ma14092374_

Round 1

Reviewer 1 Report

Dear authors please find attached my suggestions, sincerely yours,

Author Response

The authors thank the reviewer for the remarks and important inputs for revision of the manuscript. We too agree with the reviewer that the motivation for the imaging studies of conventional solidification needed to be presented with better clarity. As a result, the earlier sections have been modified to highlight the relevant purpose of experimentation (pages 7, 9 and 11 of the revised manuscript).

We wish to state that the purpose of including studies on semi-solid deformation is to complete the discussion on classical solidification defects such as hot-cracking and hot-tearing, in addition to shrinkage and gas porosity. Further, the focus of the manuscript is solidification processes in general, and is not limited to crystal growth (or solid-liquid phase transformation) alone. Therefore, we felt that it was important to include such studies, which shed light on shear bands in die cast components, combined shrinkage and hydrogen porosity in inter-granular spaces created by deformation of the semi-solid.

Reviewer 2 Report

Please double the English language before publication. The present review is good enough for publication.

Author Response

Authors thank the reviewer for their input, the manuscript has now been carefully proofread.

Reviewer 3 Report

  1. Suggest to include the limitations of applying the techniques to study different alloys? Any limitation to be used for ferrous materials? higher melting point materials. This can better help the readers to identify if the techniques apply to their research.
  2. Would the technique has sufficient resolution to study the nucleation process on the scale of ~100 nm? 

Author Response

  1. Suggest to include the limitations of applying the techniques to study different alloys? Any limitation to be used for ferrous materials? higher melting point materials. This can better help the readers to identify if the techniques apply to their research.

Response: Pages 11-12 present an overview of the high temperature solidification processes. The section is modified to bring out key challenges of experimenting with ferrous alloys more clearly.

  1. Would the technique has sufficient resolution to study the nucleation process on the scale of ~100 nm? 

Response: This is an important question. It is now possible to use imaging beamlines at APS, ESRF and DLS to perform nano-scale X-ray imaging of microstructure at ~100 nm. The nucleation events, if sufficient contrast can be obtained, can certainly be captured. However, the instrumentation for capturing these phenomena at exceptionally fast time-scales (~ 100 ns) are yet to be established. We wish to state that alternate methods such as diffraction [1] and atomic electron microscopy [2] have been used to study the nucleation kinetics.

[1] Kelton et al., First X-Ray Scattering Studies on Electrostatically Levitated Metallic Liquids: Demonstrated Influence of Local Icosahedral Order on the Nucleation Barrier, Phys. Rev. Lett. 90, 195504 –2003

[2] Scott, M., Chen, CC., Mecklenburg, M. et al. Electron tomography at 2.4-ångström resolution. Nature 483, 444–447 (2012). https://doi.org/10.1038/nature10934

Reviewer 4 Report

Paper: Progress on in situ and operando X-ray imaging of solidification processes is an excellent work in a new experimental field with many applications in metallurgy domain and other production areas.

The authors must renumber the sub-chapters in the right order - at this moment are all 1, 1.1 .... on line 29 use a dot after ) and on line 26 you can delete , after electrical

In the last section of the paper Concluding remarks the authors can give a most important perspective or further work of this technique. 

Author Response

We thank the reviewer for their constructive feedback. We apologize for the erroneous numbering, which is corrected in the revised manuscript. Similarly, the concluding remarks is updated with appropriate perspectives of the technique.